# CPRR: A Diagnostic Probe for Measuring 'Confident Uncertainty' in Large Language Model Reasoning

## Abstract

Understanding the internal reasoning processes of Large Language Models (LLMs) when confronted with complex challenges represents a core problem for interpretability research. This paper introduces a novel diagnostic probe—the Conditional Pivotal Reasoning Ratio (CPRR)—to reveal a fundamental characteristic of LLM reasoning dynamics. CPRR captures a phenomenon we term "confident uncertainty" by quantifying the model's propensity to engage in statistically surprising (high-perplexity) exploration when making high-confidence decisions.

Through an analysis of tens of thousands of reasoning paths from two LLMs with distinct training histories on the AIME mathematical competition dataset, we identify a robust pattern: on problems the models find difficult, successful reasoning paths exhibit significantly higher CPRR during the crucial initial planning phase than do failing paths. This "peak in thinking" is absent in simpler problems. The existence of this quantifiable probabilistic "signature" reveals that effective reasoning begins with a more intense initial exploration. We further substantiate through qualitative analysis that the high-frequency tokens identified by CPRR semantically constitute a "cognitive toolkit" for solving difficult problems.

This research provides a new analytical dimension for understanding how LLMs "think," shifting the focus of inquiry from the static correctness of final answers to an analysis of the dynamic, and at times noisy, reasoning process itself.

## 1 Introduction

The success of Large Language Models (LLMs) on multi-step reasoning tasks is typically measured by the accuracy of their final answers Bilal et al. (2025). However, as the field matures, this evaluation paradigm proves insufficient. It treats the reasoning process as an opaque "black box," failing to explain the critical dynamics that determine why some reasoning paths succeed while others fail. To illuminate these internal mechanisms, the research community has increasingly turned to Uncertainty Quantification (UQ) Atf et al. (2025), aiming to assess the reliability of a model's reasoning by analyzing its output probabilities Wang et al. (2025b); Liu et al. (2025).

Yet, the dominant trajectory of UQ research is distinctly performance-oriented: developing metrics that predict final accuracy, thereby enabling risk mitigation in high-stakes domains such as healthcare and law. This paper adopts a different perspective. Our goal is not to design a stronger predictor, but rather to create a diagnostic tool that reveals and characterizes the dynamic behaviors within the LLM's reasoning process. We shift the central question from "Is the final answer correct?" to "What probabilistic patterns define an effective reasoning process as it unfolds?" Bello et al. (2025)

This diagnostic lens is particularly salient in light of two major themes in current LLM research. The first concerns the phenomenon of overthinking Sui et al. (2025), where powerful reasoning models generate unnecessarily long and computationally expensive reasoning chains—even for simple problems Wei et al. (2022). This raises a fundamental question about cognitive efficiency: is more "thinking" always better? As an alternative framework, we draw inspiration from dual-process theory in human cognition, which distinguishes between a fast, intuitive, and automatic System 1 and

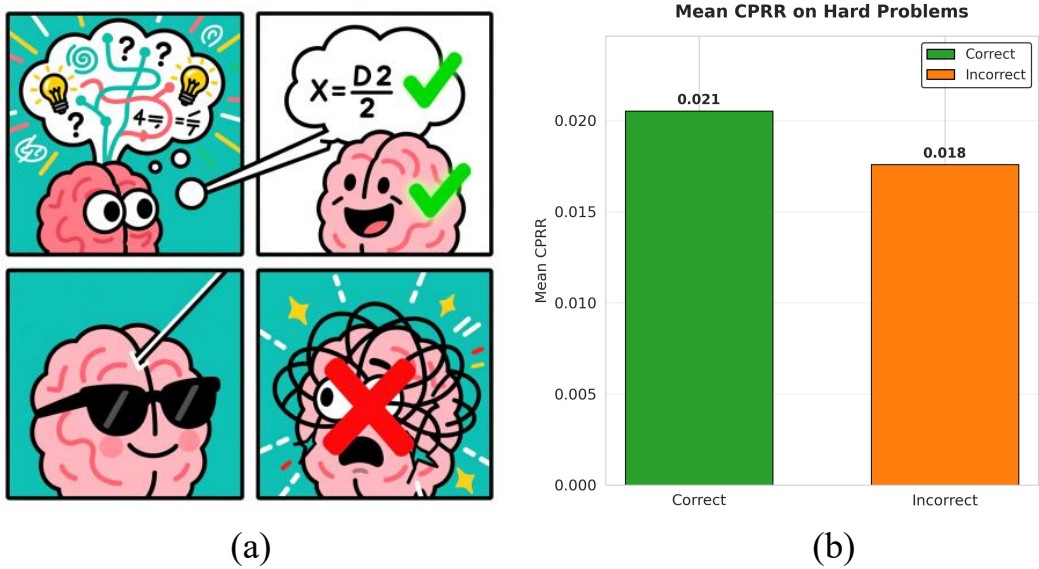

(a)                                                                    (b)

Figure 1: Reasoning patterns and their quantitative signature on hard problems. (a) **Intuitive illustration of reasoning patterns:** When tackling hard problems, successful paths (top) often begin with strategic exploration, where cognitive effort is invested to deconstruct the problem. In contrast, failed paths (bottom) can result from **overconfidence** or ineffective thinking without sufficient exploration. (b) **Quantitative evidence from CPRR:** The bar chart shows data from the 'DeepSeek-R1-0528-Qwen3-8B' model on hard problems from the AIME24 and AIME25 datasets. Successful reasoning paths (Correct) exhibit a significantly higher mean CPRR of 0.021 compared to 0.018 for unsuccessful paths (Incorrect). This suggests that a higher CPRR serves as a quantitative signature of effective, exploratory reasoning.

a slow, deliberate, and analytical System 2 Frankish (2010); Kannengiesser & Gero (2019). This theory posits that effective intelligence lies not in uniform processing, but in the ability to adaptively allocate cognitive resources based on task difficulty. As visually contrasted in Figure 1, successful reasoning strategies are context-dependent. For hard problems (a), success requires strategic exploration, while a path of premature overconfidence fails. For easy problems (b), direct thinking leads to the correct answer, whereas overthinking introduces errors and leads to failure. Our work investigates whether LLMs exhibit a functional analogue of this adaptive strategy.

To probe this, we build upon recent findings that high-entropy or statistically surprising tokens play a pivotal role in complex reasoning Wang et al. (2025a). We isolate and examine a specific probabilistic state we term confident uncertainty: moments when the model makes a statistically surprising (high-perplexity) choice while simultaneously expressing high relative certainty in that choice. We hypothesize that this state marks a non-trivial, high-information logical leap—an allocation of what can be functionally interpreted as computational effort—serving as an operational signature of System 2–like processing.

To detect this phenomenon, we introduce the Conditional Pivotal Reasoning Ratio (CPRR). Through large-scale experiments on two distinct models using the challenging AIME benchmark dataset, we identify a significant early signal: on difficult problems, successful reasoning trajectories begin with a higher proportion of effective exploration, characterized by an elevated CPRR peak—"a peak in thinking." Crucially, this peak is absent in simple problems, where the model defaults to a more direct, low-CPRR mode. The existence of this quantifiable yet subtle probabilistic signature provides the first direct evidence that effective reasoning in LLMs begins with an adaptive, high-intensity initial exploration, directly countering the narrative of indiscriminate overthinking.

Our contributions are threefold:

• Proposing CPRR as a diagnostic probe: We introduce a computationally lightweight and interpretable metric to reveal the dynamic characteristics of the LLM reasoning process.

• Identifying an early probabilistic signature: We are the first to empirically demonstrate that, on difficult problems, successful reasoning paths exhibit a distinct and quantifiable probabilistic pattern in their initial stages—absent in failed paths and in simple problems. This finding holds across models with different training backgrounds, suggesting its potential universality.

• Connecting probabilistic signals to semantics: Through qualitative analysis, we link the abstract CPRR signal to concrete cognitive functions essential for problem-solving (e.g., defining concepts, formulating hypotheses, applying principles), thereby providing a semantic foundation for interpreting the model's operational state.

## 2 RELATED WORK

Existing work can be broadly categorized into two lines of research.

**Performance-Oriented Uncertainty Quantification (UQ).** A significant body of work focuses on developing UQ metrics that accurately predict reasoning errors or improve downstream task performance. Examples include perturbation-based approaches (e.g., TOKUR Zhang et al. (2025)) and methods leveraging semantic or structural features (e.g., CoT-UQ Zhang & Zhang (2025), UQAC Li et al. (2025)). While effective in prediction, these methods often incur high computational costs or rely on complex post-processing pipelines. In contrast, CPRR does not aim to outperform them in accuracy but instead offers a complementary diagnostic perspective—one that emphasizes understanding the dynamics of the reasoning process rather than predicting final outcomes Xiong et al. (2024); Savage et al. (2024).

**Probing Cognitive and Reasoning Dynamics.** This line of research seeks to uncover the internal mechanisms underlying LLM reasoning. Our work builds upon two recent empirical observations: (1) high-entropy or statistically surprising tokens have been shown to play a critical role in effective reasoning Wang et al. (2025a); and (2) correct reasoning trajectories tend to exhibit higher overall confidence compared to incorrect ones Fu et al. (2025); Taubenfeld et al. (2025). Our key innovation lies in synthesizing these insights by specifically examining moments of *confident uncertainty*—where high surprise coincides with high relative confidence—and using this combination to identify pivotal transitions in the reasoning chain.

Furthermore, there is growing interest in proxy measures for "cognitive effort," as well as in metacognition and adaptive computation inspired by cognitive science. Notably, the phenomenon of *overthinking*—where LLMs generate unnecessarily long reasoning chains even for simple problems—has recently drawn widespread attention, raising concerns about computational inefficiency. Our findings are directly relevant to this discussion and provide quantitative counterevidence: the "peak in thinking" captured by CPRR emerges only during the initial phase of solving difficult problems and vanishes on simple ones. This suggests that effective reasoning is not characterized by indiscriminate lengthening of thought, but by an adaptive allocation of computational resources based on perceived task difficulty—a finding that offers concrete, measurable support against blind overthinking Vamvourellis & Mehta (2025); Sui et al. (2025).

In sum, CPRR's value lies in being a computationally efficient and conceptually simple diagnostic probe that reveals previously unobserved temporal dynamics associated with effective computational effort. It provides a window into the model's early-stage reasoning state—an aspect largely overlooked by conventional UQ methods focused on end-state predictions.

## 3 METHODOLOGY

### 3.1 CORE METRIC DEFINITIONS

Our analysis is built upon two token-level metrics—perplexity and confidence—designed to capture distinct aspects of model behavior during reasoning: statistical surprise and decisional decisiveness.

**Perplexity (PPL):** For a generated token $t_i$, its perplexity is defined as the exponential of the negative log-probability:

$$\text{PPL}(t_i) = \exp(-\log P(t_i \mid t_{<i})) \tag{1}$$

Higher PPL indicates that the token is statistically unexpected given the context, reflecting a non-trivial or surprising transition in the reasoning trajectory.

**Confidence:** The concept of using model confidence to evaluate reasoning has been explored in prior work, where studies have shown that correct reasoning paths often exhibit higher confidence Fu et al. (2025). To measure a dimension orthogonal to PPL—namely, decisional decisiveness—we avoid using the sampled token's raw probability, which is inherently and strongly negatively correlated with PPL. Instead, we define confidence as the relative advantage of the selected token over its competitors. Specifically, it is computed as the average negative log-probability of all non-sampled candidate tokens at that step:

$$\text{Confidence}(t_i) = -\frac{1}{|V \setminus \{t_i\}|} \sum_{t' \in V \setminus \{t_i\}} \log P(t' \mid t_{<i}) \tag{2}$$

where $V$ represents the sampling pool of candidate tokens, not the entire vocabulary. In our experiments, we employ a top-k sampling strategy where $k = 20$, thus $|V| = 20$. A higher confidence score indicates that the chosen token $t_i$ dominates the distribution over the candidate pool—even if its absolute probability is low—as long as it is substantially more likely than the alternatives.

This non-standard definition serves a critical analytical purpose: to orthogonally decouple statistical surprise from decisional certainty. Using standard confidence (i.e., $P(t_i)$) would make high-PPL and high-confidence co-occurrence nearly impossible due to their inverse relationship. In contrast, our metric captures the "peakedness" of the output distribution, enabling us to identify moments when the model makes a surprising yet decisive choice—the hallmark of our proposed confident uncertainty state.

### 3.2 CONDITIONAL PIVOTAL REASONING RATIO (CPRR)

We define a Pivotal Reasoning Token (PRT) as any token that satisfies both: High Perplexity and High Confidence.

To quantify the extent to which a model engages in such non-trivial, confident exploration during reasoning, we introduce the Conditional Pivotal Reasoning Ratio (CPRR):

$$\text{CPRR} = \frac{(t_i \in \text{High PPL} \cap \text{High Conf})}{(t_i \in \text{High Conf})} \tag{3}$$

This ratio measures the proportion of decisive generation steps that also involve a statistically surprising leap. In essence, CPRR operationalizes the intensity of effective exploratory effort—a signature of deliberate, System 2–like processing—within the reasoning process.

### 3.3 EXPERIMENTAL SETUP

**Models:** We evaluate two variants derived from the Qwen3-8B architecture but trained under different paradigms:

(1) **Qwen3-8B** Team (2025): A general-purpose model developed by Alibaba Cloud via knowledge distillation from a larger teacher, featuring strong baseline reasoning capabilities;

(2) **DeepSeek-R1-0528-Qwen3-8B** (R1-Qwen3-8B) DeepSeek-AI (2025): A reasoning-specialized variant obtained by post-training on `Qwen3-8B-base` using chain-of-thought trajectories distilled from the proprietary *DeepSeek-R1-0528* model DeepSeek-AI (2025), with reinforcement learning–guided alignment to enhance logical coherence.

Despite architectural similarity, the two models differ in their training data, objectives, and reasoning optimization strategies—enabling a controlled comparison of probabilistic reasoning dynamics.

**Dataset:** We use AIME2024 and AIME2025, widely recognized benchmarks for evaluating complex mathematical reasoning in LLMs. These datasets feature challenging problems across algebra, combinatorics, and geometry, providing a rigorous testbed for multi-step logical inference Balunović et al. (2025).

**Sampling Strategy:** For each question and model, we generate 64 independent reasoning trajectories. All generations use a consistent decoding configuration: a temperature of 0.6, top-p sampling

of 0.95, and top-k sampling of 20. The maximum number of generated tokens is set to 32,768 to ensure sufficient coverage of diverse reasoning paths.

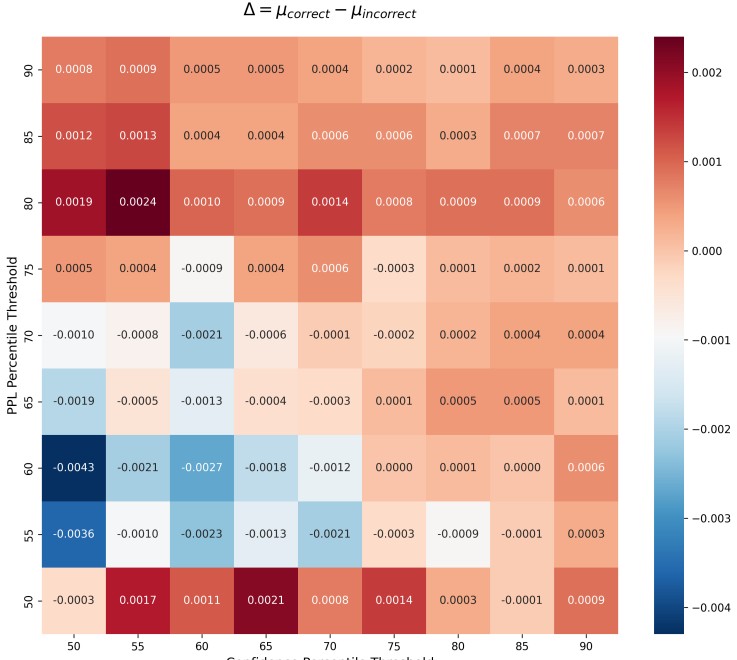

Figure 2: Sensitivity Analysis Heatmap for Determining Optimal Thresholds. This analysis was conducted to empirically derive an optimal and generalizable set of thresholds for our methodology. We performed this detailed sensitivity analysis on the 'R1-Qwen3-8B' model using hard problems from the AIME25 dataset as a representative case. The heatmap displays the difference in mean CPRR between correct and incorrect paths ($\Delta = \mu_{\text{correct}} - \mu_{\text{incorrect}}$) across a grid of PPL (y-axis) and Confidence (x-axis) percentile thresholds.

### 3.4 ANALYSIS FRAMEWORK

Our analysis framework is designed to ensure robust and reliable findings. We stratify problems by difficulty into hard ($<25\%$ model accuracy) and easy ($>75\%$ accuracy) categories. To ensure statistical rigor, all reported mean differences are accompanied by 95% confidence intervals estimated via bootstrap resampling ($N = 1000$).

A critical aspect of our methodology is the selection of thresholds for defining a Pivotal Reasoning Token. To derive an empirically optimal and generalizable set of thresholds, we conducted a systematic sensitivity analysis on the R1-Qwen3-8B model using hard problems from the AIME25 dataset as a representative case. To focus our analysis on the core reasoning phase and mitigate statistical noise, we applied two filtering criteria to the token data. First, we limited the tokens involved in threshold and CPRR calculations to the initial 90% of each reasoning process. Second, to reduce interference from extreme PPL values caused by stochastic sampling, we excluded any token with a PPL greater than 100. We then performed a grid search across a range of percentile thresholds for both PPL and Confidence, computing the difference in mean CPRR between correct and incorrect reasoning paths ($\Delta = \mu_{\text{correct}} - \mu_{\text{incorrect}}$) for each pair.

The results of this analysis, visualized in the heatmap in Figure 2, reveal a clear and crucial pattern: the desired positive effect ($\Delta > 0$), where successful paths have a higher CPRR, is robustly concentrated in the region where the PPL threshold is high (e.g., above the 75th percentile). This finding aligns with our intuition: while many tokens may be generated with high relative confidence, only the most statistically surprising (high PPL) tokens mark truly pivotal steps in complex reasoning, thus requiring a higher PPL threshold to effectively capture the signal.

Based on this data-driven evidence from the AIME25 analysis, we selected the 80th percentile for PPL and the 55th percentile for Confidence as our fixed, global thresholds. As shown in the heatmap,

this combination ($\Delta = 0.0024$) corresponds to one of the strongest and most stable points within the effective region. By deriving these thresholds from a single representative dataset and applying them universally, we can rigorously test the generalization of our core findings across all other experimental conditions.

## 3.5 CORE FINDING: HARD PROBLEMS REQUIRE A HIGHER-INTENSITY "PEAK IN THINKING"

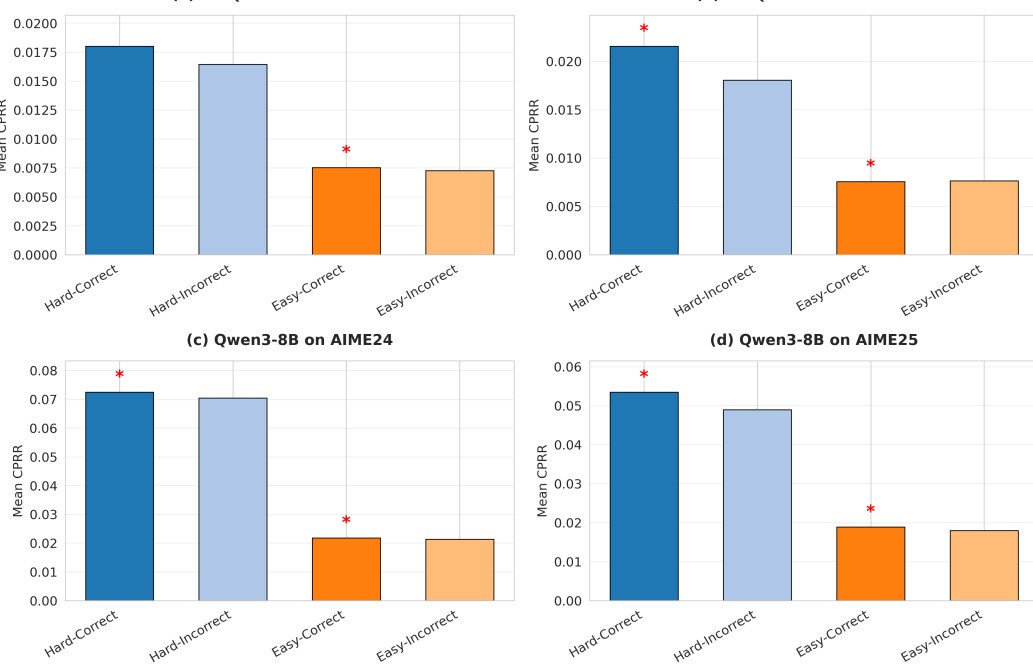

Figure 3: Global comparison of mean total Conditional Pivotal Reasoning Ratio (CPRR) across problem difficulty and path correctness. The analysis spans two models, (a, b) DeepSeek-R1-0528-Qwen3-8B and (c, d) Qwen3-8B, on the AIME24 and AIME25 datasets. A robust "thinking peak" pattern is observed: successful reasoning paths on hard problems (Hard-Correct) consistently demonstrate a significantly higher CPRR than unsuccessful paths (Hard-Incorrect) and all paths for easy problems. This effect is notably absent in easy problems, suggesting that the model adaptively allocates exploratory effort. Red asterisks (*) denote a statistically significant difference ($p < 0.05$) between correct and incorrect paths for a given difficulty.

## 4 RESULTS

Our central finding is that CPRR serves not only as a reliable indicator of effective reasoning but also that its intensity is adaptively modulated based on perceived problem difficulty. As shown in Figure 3, a highly consistent pattern emerges across all four experimental settings—spanning two models (Qwen3-8B and DeepSeek-R1-0528-Qwen3-8B) and two datasets (AIME24 and AIME25):

First, CPRR acts as a universal indicator of effective reasoning regardless of difficulty. For both hard and easy problems, successfully solved reasoning paths (Hard-Correct, Easy-Correct) consistently exhibit a significantly higher mean CPRR in their early phase compared to their failed counterparts (Hard-Incorrect, Easy-Incorrect). As denoted by the asterisks, this difference is statistically significant in nearly all cases.

However, our most crucial finding is that the required "intensity of thinking" for success is directly correlated with problem difficulty. The magnitude of CPRR for hard problems is substantially higher than for easy problems. For instance, across both models and both datasets, the mean CPRR for

successful paths on hard problems was consistently and substantially higher than for successful paths on easy problems. This provides strong evidence that the model allocates a much higher intensity of exploratory thinking (manifested as a higher CPRR) when faced with a greater challenge. This dual finding provides strong evidence for the functional specificity of CPRR: it not only reflects answer correctness but also captures the intensity of quantifiable "computational effort" invested to overcome cognitive challenges. This result also offers direct counterevidence to indiscriminate overthinking: effective reasoning lies not in longer chains, but in the ability to initiate high-intensity, adaptive exploration when needed.

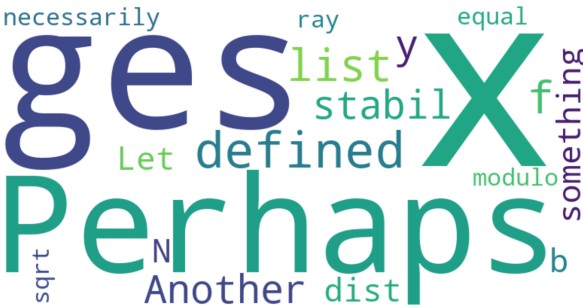

Figure 4: Word cloud of high-frequency Pivotal Reasoning Tokens (PRTs) from successful reasoning paths on hard problems.

## 4.1 QUALITATIVE ANALYSIS: THE "COGNITIVE TOOLKIT" OF PIVOTAL REASONING TOKENS

To understand the semantic meaning behind the CPRR signal, we conducted a qualitative analysis of Pivotal Reasoning Tokens (PRTs) in successful paths on hard problems (see Figure 4). These tokens are not randomly distributed but systematically form a **"cognitive toolkit"** essential for solving complex mathematical tasks. We categorize them by functional role:

**Problem Framing and Hypothesis Setting:** Tokens like `Let`, `defined`, `assume`, and `Perhaps` mark foundational steps—defining variables, clarifying concepts, or proposing exploratory hypotheses.

**Core Mathematical Concept Application:** Tokens such as `modulo`, `divisible`, `finite`, `pairs`, and `symmetric` indicate the activation of domain-specific knowledge and abstract structures.

**Logical Operations and Assertions:** Tokens like `equal`, `implies`, `solutions`, and `exactly` represent critical junctures where logical progress is made.

The systematic emergence of this toolkit supports our claim that CPRR reflects genuine, structured, and goal-directed higher-order cognition—not just statistical noise.

## 4.2 ROBUSTNESS CHECK: SIGNAL REFINEMENT AT INCREASED SAMPLING SCALE

To verify that our findings are not an artifact of limited sampling, we conducted an expanded sampling experiment. For a set of hard problems from the AIME25 dataset, we increased the number of reasoning trajectories per question from 64 to 256 for the R1-Qwen3-8B model.

Interestingly, we observed that at this larger sampling scale, the CPRR difference between correct and incorrect paths became less pronounced using our original global thresholds (PPL 80, Conf 55). We posit that this is because 256 trajectories cover a vastly wider and more diverse set of reasoning strategies than 64, introducing more "mediocre" or "atypical" paths that add noise and dilute the original signal.

We hypothesized that within such a large and diverse sample space, the truly "elite" pivotal reasoning steps that distinguish success from failure become sparser and thus require a more stringent defini-

tion to be detected. To test this, we re-analyzed the data using higher thresholds (90th percentile for PPL and 65th for Confidence) to isolate only the most exceptionally surprising and decisive tokens.

As illustrated in Figure 5, the "thinking peak" phenomenon re-emerged with clarity under these more stringent thresholds. For a majority of the examined problems, the mean initial CPRR for correct paths was once again substantially higher than for incorrect paths. This result is significant for two reasons. First, it confirms that the "thinking peak" is a real and robust phenomenon, not a small-sample artifact. Second, it provides a deeper insight into the nature of the CPRR signal: as the diversity of explored paths increases, the decisive logical leaps that separate success from failure become rarer and more critical, thus requiring a higher bar for their identification. This demonstrates both the stability and the nuanced character of our findings.

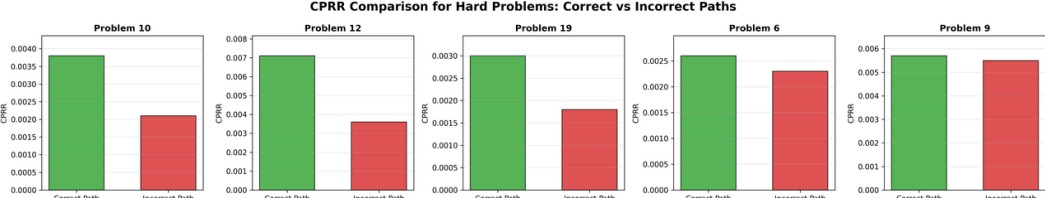

Figure 5: Re-emergence of the 'thinking peak' under stricter analysis. The CPRR of correct vs. incorrect paths is shown for five hard AIME25 problems using the R1-Qwen3-8B model with 256 samples. The signal is refined using higher thresholds (90th percentile PPL, 65th percentile Confidence), confirming the phenomenon's robustness.

## 5 DISCUSSION

Our study uncovers a subtle yet profound dynamic in how large language models (LLMs) tackle complex problems. CPRR serves as a diagnostic probe, offering a functional window into the allocation of what can be interpreted as "computational effort" during reasoning.

### 5.1 FROM EARLY SIGNAL TO PROBABILISTIC "SIGNATURE"

The core finding is that successful reasoning leaves a detectable probabilistic "signature" in the model's output. This signature manifests as a higher proportion of confident uncertainty—defined as high-surprise yet decisive token choices—and emerges within the first 10% of the reasoning trajectory. This pattern is consistently observed across both the Qwen3-8B and R1-Qwen3-8B, suggesting that this dynamic is not an artifact of specific training procedures but may instead reflect a more fundamental property of effective generative reasoning.

### 5.2 CPRR AS A PROXY FOR "EFFECTIVE COMPUTATIONAL EFFORT" AND ALIGNMENT WITH DUAL-PROCESS THEORY

Our findings exhibit a striking functional correspondence with the well-established dual-process theory in cognitive science, which distinguishes between two modes of thinking: fast, intuitive, automatic System 1, and slow, deliberate, resource-intensive System 2. Within our framework Kannengiesser & Gero (2019):

The low-CPRR mode exhibited on easy problems functionally resembles efficient System 1 processing—enabling rapid responses with minimal computational overhead. In contrast, the "peak in thinking" characterized by high CPRR in successful paths on hard problems aligns precisely with the activation of System 2: a deliberate, exploratory, and cognitively intensive process necessary for solving non-trivial challenges.

Thus, CPRR transcends being merely a statistical metric; it acts as a functional detector, providing the first clear quantitative evidence that LLMs may exhibit human-like, task-adaptive switching between cognitive strategies. On hard problems, elevated early CPRR can be interpreted as increased investment in effective computational effort for exploration and planning. The absence of this pattern on easy problems directly challenges the prevailing narrative that LLMs indiscriminately en-

gage in "overthinking". Instead, our results suggest that models can adaptively reduce unnecessary computation, offering strong evidence for intelligent, demand-driven resource allocation.

### 5.3 IMPLICATIONS FOR META-REASONING AND FUTURE APPLICATIONS

While insufficient for a robust classifier, CPRR's significance lies not in predictive power but in revealing the nature of the signal itself: a real yet noisy early warning. This is precisely the kind of input required by systems capable of meta-reasoning or self-correction.

For an agent aiming to monitor and improve its own reasoning, a perfect signal arriving only at the end of the chain is useless for real-time intervention. In contrast, a noisy but directional early signal holds immense strategic value. For instance, a more advanced LLM could monitor its initial CPRR when encountering a problem it deems difficult. If CPRR is anomalously low, it could trigger a "self-reflection" mechanism—such as re-examining the question, switching strategies, or allocating additional computational resources for broader search.

Therefore, while CPRR is not a perfect predictor, its value as an actionable early diagnostic signal makes it a promising building block for next-generation intelligent architectures. It has the potential to become a key input for developing LLMs with greater strategic awareness, adaptability, and robustness Liao & Varshney (2021).

## 6 CONCLUSION

This work introduces the Conditional Pivotal Reasoning Ratio (CPRR) as a diagnostic probe to dissect the reasoning dynamics of large language models. Rather than pursuing a performance-optimized UQ metric, we identify and validate a more fundamental phenomenon: successful reasoning paths exhibit a distinct probabilistic "signature"—characterized by a higher proportion of confident uncertainty—at remarkably early stages of the generation process.

This finding is substantiated through large-scale experiments across two models with different training histories, qualitative semantic analysis, and extensive robustness checks. Our primary contribution lies in offering a new analytical lens for LLM interpretability—one that shifts focus from whether a model gets the right answer to how it arrives at that answer at a procedural level. We hope CPRR will inspire further investigation into the internal mechanisms of LLM reasoning and lay the groundwork for building next-generation intelligent systems with built-in monitoring and adaptive capabilities.

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

## A   APPENDIX

### LARGE LANGUAGE MODEL USAGE DISCLOSURE

This work involved the use of large language models (LLMs) during the preparation process. Specifically, an LLM was used to assist with language editing and improving the clarity of the manuscript. Additionally, an LLM was utilized to help generate and refine experimental code. Following the ICLR 2026 policies, we disclose this usage and affirm that all final content, results, and claims are the responsibility of the human authors.

