# OpenReview forum: "CPRR: A Diagnostic Probe for Measuring 'Confident Uncertainty' in Large Language Model Reasoning"
_ICLR.cc/2026/Conference — ICLR 2026 Conference Withdrawn Submission_

### Official Review · Reviewer_hAMV · 2025-10-27

**Soundness:** 2
**Presentation:** 2
**Contribution:** 2
**Rating:** 2
**Confidence:** 4

**Summary:**

This paper proposes a targeted diagnostic probe called the Conditional Pivotal Reasoning Ratio (CPRR), designed to capture the phenomenon where a large language model engages in high-perplexity exploration even when making high-confidence decisions. The authors refer to this behavior as confident uncertainty. Based on the AIME mathematics competition dataset, they collect reasoning trajectories from two language models with different training histories. They find that for problems the models consider difficult, successful reasoning paths exhibit significantly higher CPRR during the initial planning phase compared to failed paths. This "peak in thinking" is not observed in easier problems.

**Strengths:**

- Clear Motivation
- Contains a Disclosure of LLM Usage

**Weaknesses:**

- Poor Writing
- Lack of Technical Contribution
- Insufficient Experiments and Analysis

**Questions:**

I have read this paper very carefully. Although it presents a clear motivation, I believe the authors must address the following issues before the paper can be considered for publication.

1. The description in the Method section is too limited, and I do not understand why confidence is defined over a top-k=20 candidate set. The authors provide neither experimental justification nor explanation for this choice, which makes me seriously question the validity of the conclusions. There are also other arbitrary hyperparameter choices that are not explained, such as discarding the last 10% of tokens and filtering out tokens with perplexity greater than 100.
2. The main claim of the paper is that a peak appears during the initial phase of reasoning. However, I do not see any clear quantitative result to support this, such as stating that a peak consistently occurs within the first x% of tokens. Moreover, the observed peak difference does not seem substantial from the results shown.
3. The study includes only two models (Qwen3-8B and DeepSeek-R1-0528-Qwen3-8B) and two datasets (AIME2024 and AIME2025). There is a lack of experiments across tasks and model scales. This makes it difficult to assess the generality of the findings. Therefore, I find the claim of a "robust pattern" in the abstract to be overstated.
4. It seems the paper measures token-level perplexity on the model’s own generated sequences. But the sampling strategy clearly influences which tokens are selected, which in turn affects the observed outcomes. However, the decoding configuration is fixed throughout (T=0.6, top-p=0.95, top-k=20), and no sensitivity analysis is conducted on sampling strategies.
5. My biggest concern is that the paper appears to simply report an observed phenomenon (termed "confident uncertainty") without providing a deep analysis. While the phenomenon seems related to reasoning success or failure, the paper mainly presents correlations between different metrics without further investigation. This limits the contribution of the work.

In summary, the paper lacks sufficient experimentation and analysis, and the writing has many issues that need to be corrected.

Minors:

1. There are many formatting errors with quotation marks. For example, in line 17, the quotation marks around "confident uncertainty" are both downward facing, which is incorrect.
2. Abbreviation usage is inconsistent. For instance, “Large Language Models (LLMs)” appears four times throughout the paper, which is unnecessary.
3. Equation (1) lacks a period at the end, and Equation (2) is missing a comma.
4. The in-text citation format is incorrect throughout the paper.

---

### Official Review · Reviewer_d3y4 · 2025-10-29

**Soundness:** 1
**Presentation:** 2
**Contribution:** 2
**Rating:** 2
**Confidence:** 3

**Summary:**

This paper introduces a novel diagnostic tool—the Conditional Pivotal Reasoning Ratio (CPRR)—to uncover the dynamic characteristics of LLMs when reasoning through complex problems. By analyzing the reasoning paths of two LLMs with distinct training histories on the AIME mathematical competition dataset, the study finds that successful reasoning paths exhibit significantly higher CPRR during the initial planning phase for difficult problems, whereas this "peak in thinking" is absent in simpler problems.

**Strengths:**

The paper introduces CPRR as a new metric, providing a fresh perspective for understanding the reasoning processes of LLMs. The experimental sensitivity analysis adds robustness to the findings, which contribute valuable insights into the mechanisms of LLM reasoning.

**Weaknesses:**

Lack of Detailed Comparison with Existing Methods: Although the paper mentions differences from other UQ methods, it does not provide a detailed comparison of the specific advantages and disadvantages of CPRR versus existing approaches.

Insufficient Experimental Evidence: The conclusions drawn are based on only two models and two datasets, which may not be sufficient to generalize the findings.

Lack of Detail in Overall Paper: The paper lacks detailed support for its claimed contributions, such as "Connecting probabilistic signals to semantics" and the discussions in Section 5, which reduces the persuasiveness of the arguments.

**Questions:**

Applicability of CPRR: The use of only two datasets and two models raises concerns about the general applicability of CPRR. Drawing conclusions like "Our findings exhibit a striking functional correspondence with the well-established dual-process theory in cognitive science" based solely on mathematical problems seems premature.

The paper cited in Section 5 mentions, "Recently, it has been understood as a concept encompassing a set of techniques aimed at stimulating creative solutions to problems in business, education, and social domains" (Kannengiesser & Gero, 2019).
Intuitively, equating creative solutions with mathematical problem-solving appears problematic and warrants further clarification.

---

### Official Review · Reviewer_5eAW · 2025-11-01

**Soundness:** 3
**Presentation:** 2
**Contribution:** 2
**Rating:** 2
**Confidence:** 4

**Summary:**

This paper introduces the Conditional Pivotal Reasoning Ratio (CPRR), a novel diagnostic probe for analyzing the internal reasoning dynamics of large language models (LLMs). CPRR captures where LLMs make statistically surprising (high-perplexity) yet high-confidence token choices during reasoning. The authors analyze two LLMs on AIME datasets, finding that successful reasoning paths on difficult problems exhibit a distinct early "thinking peak" absent in simpler problems and in failed reasoning paths. This suggests adaptive allocation of computational effort analogous to human dual-process theory's System 1 and System 2 thinking. The study also qualitatively links high-CPRR tokens to semantic cognitive functions essential for problem-solving.

**Strengths:**

The paper introduces CPRR, a creative and interpretable diagnostic metric that aims to quantify "confident uncertainty" in LLM reasoning.

The paper does not rely solely on probabilistic metrics. It adds a valuable layer of interpretability by qualitatively analyzing the high-CPRR tokens, linking them to a semantic "cognitive toolkit".

**Weaknesses:**

The evaluation is limited to mathematical reasoning benchmarks (AIME dataset), leaving questions about the generalizability of CPRR to other reasoning domains, such as commonsense reasoning.

The stratification of easy and hard problems is based on model accuracy within the AIME dataset, a benchmark known for "complex reasoning problems". Therefore, the easy problems are not genuinely simple tasks; they are the least difficult of an intrinsically complex set. So it makes it unclear if the low-CPRR mode observed on easy problems represents a true System 1 response, or just a more efficient and well-practiced System 2 path. Because all problems are inherently complex, the study likely just contrasts two modes of analytical reasoning rather than demonstrating a true switch between intuitive and analytical cognition. The comparison should be with AIME and other datasets of elementary or mid-level mathematical problems (e.g., GSM8K).

In Section 4.2, the authors report that when the number of reasoning trajectories was increased from 64 to 256, the CPRR difference between correct and incorrect paths became less pronounced. They attribute this to added noise from more "mediocre" or "atypical" reasoning paths, but instead of systematically analyzing this result, they modified their thresholds post hoc (raising the PPL and Confidence percentiles) until the thinking peak pattern reappeared. This weakens the robustness claim. It suggests that the CPRR signal is not stable under scaling and may depend on selective tuning rather than reflecting a consistent, intrinsic property of reasoning dynamics.

**Questions:**

Do you expect the same "thinking peak" phenomenon on non-mathematical reasoning tasks (e.g., commonsense) to generalize?

Given that all AIME problems involve complex reasoning, isn’t it possible that the "easy" vs. "hard" contrast reflects two levels or modes of analytical reasoning rather than a genuine switch between intuitive (System 1) and analytical (System 2) cognition? How do the authors justify interpreting this contrast as evidence of dual-process behavior instead of levels of analytical reasoning?

When CPRR differences vanish at larger scales, could that indicate that reasoning efficiency is not a stable model property but rather an artifact of sampling noise?

---

### Official Review · Reviewer_xEgN · 2025-11-04

**Soundness:** 1
**Presentation:** 2
**Contribution:** 1
**Rating:** 2
**Confidence:** 3

**Summary:**

This paper introduces the Conditional Pivotal Reasoning Ratio (CPRR), a diagnostic probe designed to measure a phenomenon termed "confident uncertainty" in LLM reasoning. Confident uncertainty is operationalised as the propensity for an LLM to generate tokens that are statistically surprising (High Perplexity) while simultaneously making a high-confidence (decisive) choice among candidate tokens.

The paper evaluates the CPRR using two Qwen3-8B-based models on the AIME datasets. Their finding is that successful reasoning paths on difficult problems exhibit a higher mean CPRR during the initial phase than unsuccessful paths or paths for easy problems. Based on this pattern, which is referred to as a "peak in thinking," the paper suggests that effective reasoning in LLMs involves a high-intensity initial exploration, which is claimed to be aligned with the System 2 mode of the dual-process theory. Qualitative analysis of the high-CPRR tokens suggests a "cognitive toolkit" of concepts and logical operations crucial for problem-solving.

**Strengths:**

- The paper introduces and operationalises the concept of "confident uncertainty."

- The empirical finding that the high-CPRR mode is active for hard problems but absent for easy problems suggests that LLMs might have different patterns in reasoning process.

**Weaknesses:**

- Calling the probabilistic pattern universal as in line 110-111 is an overclaim. The entire empirical study is constrained to a single, highly specific domain: complex mathematical reasoning (AIME dataset). There is no evidence provided to suggest that the phenomenon of "confident uncertainty," generalises to other reasoning tasks, such as common sense, knowledge-intensive, or legal reasoning.

- The core definition of a PRT and consequently the CPRR depend entirely on empirically derived percentile thresholds for Perplexity and Confidence. When the sampling scale was increased from 64 to 256 trajectories, a different set of thresholds (90th Perplexity, 65th Confidence) was required to re-establish the "thinking peak". This indicates that the demonstrated signal is not robust and intrinsic.

- The observation in this paper is only tested on two LLMs. It is difficult to tell whether the results are generalisable to other models.

**Questions:**

The work is limited to the AIME mathematical domain. Is there any evidence or theoretical reasoning to suggest that the "peak in thinking" captured by CPRR would apply to non-mathematical reasoning tasks?

---

### Note · Authors · 2025-11-26

I have read and agree with the venue's withdrawal policy on behalf of myself and my co-authors.